# Effects of different fertilization modes on rice yield and quality under a rice-crab culture system

**Wanning Zhao**[1☯], **Hanling Liang**[1☯], **Yu Fu**[1☯], **Yubo Liu**[1☯], **Chen Yang**[1☯], **Tian Zhang**[1☯], **Tianyu Wang**[1☯], **Liyan Rong**[1☯], **Shuang Zhang**[1☯], **Zhaoxia Wu**[1]*, **Wentao Sun**[2]*

**1** College of Food, Shenyang Agricultural University, Liaoning Province, Shenyang, PR China, **2** Institute of Plant Nutrition and Environmental Resources, Liaoning Academy of Agricultural Sciences, Liaoning Province, Shenyang, PR China

☯ These authors contributed equally to this work.
* wuzxsau@163.com (ZW); wentaosw@163.com (WS)

**Data Availability Statement:** All relevant data have been uploaded to figshare and are accessible using the following URL: https://figshare.com/s/

## Abstract

Rice-crab culture is the characteristic rice ecological breeding model used in the Panjin area of Liaohe River Basin, China, and it can improve the ecological environment and create increased economic benefits. From a food perspective, both rice yield and quality, which are closely related to the fertilization mode, should be considered. However, the effect of different fertilization modes on rice quality has not been comprehensively investigated in this co-culture system. This study investigated the effects of three fertilization modes(FP1, FP2, and OPT) divided according to different fertilization types and methods on rice yield and quality, and set up a non-fertilized control group. In the rice-crab culture system, FP2 used fewer fertilizers and had a lower economic cost, and the yield was only slightly less than that of OPT(highest yield) but there was no statistical difference. FP2 elicited the best appearance quality and better cooking and eating quality among all treatment modes. Compared with CK, three fertilization modes significantly increased the protein content in rice and decreased the amylose content, which would lead to the deterioration of rice eating quality. However FP2 had the least protein increase and the least amylose reduction. There was no significant change in crude fat and starch content. Therefore, the FP2 fertilization mode was the best choice for the rice-crab culture system, as it significantly improved rice yield and increased rice quality at a relatively low cost.

## Introduction

Rice (*Oryza sativa L.*) is one of the three major food crops worldwide. Approximately 50% of the world's population uses rice as a staple food. Rice production in China accounts for 30% of the world's total production, ranking the first worldwide. China has a long history of rice cultivation and has formed a characteristic cultivation model that conforms to the geographical environment [1]. The Liaohe River Basin is rich in Chinese mitten crabs and is one of the three major crab producing areas in China. Panjin City is located in the southern part of the

30b0d5609f8ba6b971cf and DOI: 10.6084/m9.
figshare.11582922.

**Funding:** WZX,ZWN,LHL,FY,LYB,YC,ZT,WTY,RLY,
ZS received award of this work. This work was
supported by grants from the National Key
Research and Development Program of China
(grant no. 2018YFD0200203) and Science and
Technology Innovation Talents Training Project of
Liaoning Province (grant no. XLYC1802044). The
funders had no role in study design, data collection
and analysis, decision to publish, or preparation of
the manuscript.

**Competing interests:** The authors have declared
that no competing interests exist.

alluvial plain of the Liaohe River Basin. It is widely planted with rice and is one of the main rice-producing areas in the Liaohe River Basin [2]. Since the last century, the Panjin area has gradually developed a special cultivation mode of rice and crab co-culture, which has become an important ecological agricultural model in the Liaohe River Basin. The organic rice-crab culture not only increases rice yield, but also increases soil organic carbon content, effectively improving the quantity and composition of soil carbohydrates [3,4]. Rice and crab co-culture can control and reduce pests and weeds and decreases the use of fertilizers and pesticides [5]. This co-culture also increases crab production [6]. It not only improves the ecological environment, but also produces greater economic benefits. However, previous studies have focused on rice yield under the rice-crab culture system and research on rice quality is rare [7].

Rice quality comprises milling, appearance, cooking and eating and nutritional quality, etc. With the continuous improvement of economic life, people's requirements for rice quality, especially flavor, taste and nutritional quality, are becoming stricter[8]. Rice qualities are affected by many factors such as heredity and the environment[9]. The application of fertilizer plays a crucial role in improving rice yield and is also a key factor affecting rice quality [10]. Studies have shown that different fertilizer types, as well as fertilization modes and rates directly affect rice yield and quality [11–13]. The rice-crab culture system is no exception. In Panjin area, local farmers have accumulated more than 10 years of experience in the fertilization technology of the rice-crab culture system [14]. Three representative fertilization modes have been formed and their effects on rice yield and quality of rice-crab culture system need further study. The purpose of this test is to determine a more suitable fertilization mode and amount for producing rice under the rice-crab culture system in Panjin area based on comprehensive measurement of rice yield and quality under the above three fertilization modes.

## Materials and methods

### Materials

The test area of the present study was located at Hangcheng Farm (122°14′17″ N, 41°9′31″ E) of Panjin, Liaoning Province, China. The tested soil was saline soil. The soil parameters were tested in advance with method by Lu, et, al [15]. The soil was pH 8.20, organic matter 22.6 g/kg, total nitrogen 1.42 g/kg, available nitrogen 105.2 mg/kg, effective phosphorus 21.6 mg/kg, available potassium 164.2 mg/kg, and bulk density 1.39 g/cm$^3$. The sample used was the main rice variety Yanfeng 47 from the coastal rice-growing area of Liaoning. Chinese mitten crabs were symbiotic with this rice. The rice was transplanted on May 25, 2018 and harvested on October 8. The crab juveniles were admitted to the farm on May 28 and harvested on September 30. Each planting area was 75 m$^2$, the plot was 25 m long and 3 m wide, and the planting distance × plant spacing was 9 inches × 5 inches. Plastic spaces were built between the communities and the peripheral network of the community was used to prevent crabs from fleeing. The experimental design was randomized complete block design with three replications.

At present, there are three typical fertilization modes in rice-crab culture system in Panjin area: 1) FP1: traditional fertilization mode, 100% inorganic fertilizer, and total fertilization amount of 540 kg hm$^{-2}$, of which the ratio of nitrogen, phosphorus, and potassium was 20:7:9. Phosphorus and potassium fertilizers were applied together as the base fertilizers, and 60% of the nitrogen fertilizers were applied as base fertilizers, 30% were applied at the tillering stage, and 10% were applied at the heading stage. 2) FP2: One-time application of organic-inorganic compound fertilizer, with a total fertilization amount of 379.5 kg hm$^{-2}$, of which the ratio of nitrogen, phosphorus, and potassium was 14:5:6.3. The ratio of inorganic nitrogen to organic nitrogen was 7:3, inorganic nitrogen provided 70% of the nitrogen in the mixed fertilizer and organic nitrogen provided 30% of the nitrogen. Chicken manure was used as organic fertilizer,

and the nitrogen content was determined in advance. 3) OPT: One-time application of inorganic fertilizer, with a total fertilization amount of 465 kg hm$^{-2}$, of which the ratio of nitrogen, phosphorus, and potassium was 18:7:6. OPT fertilization mode was considered by the local farmers to be the best amount of fertilizer to apply. To date, no studies have examined the effects of these three different fertilization modes on rice yield and quality in the rice-crab culture system in the Panjin area. Therefore, the present study used rice variety Yanfeng 47 as the experimental material and set a blank control group CK without fertilization to study the effects of three different fertilization modes on rice yield, milling, appearance, cooking and eating and nutritional quality of rice in the rice-crab culture system.

Rice was harvested at maturity (8$^{th}$, Oct). After drying to a moisture content of 14%, the rice was stored at room temperature (15–30°C) and then subjected to quality measurements.

## Methods

Milling quality: was based on the GB/T 17891–1999 (1999) standard to determine the brown, milled, and head rice rates. The rice was shelled two times using a rice sheller (FC2K, Yamamoto, Japan) and rice machine (VP-32T, Yamamoto, Japan) to obtain brown rice and milled rice, respectively, and weighed separately. The head rice rate was measured using the ES-1000 appearance quality tester (Shizuoka, Japan).

Appearance quality: chalky kernel rate and chalkiness were measured by image processing measurement in GB 1354 using an ES-1000 appearance quality detector (Shizuoka, Japan).

Cooking and eating quality: the cooked rice soup was cooled to room temperature and the pH of the rice was measured with a PB-10 pH meter (German Sartorius Group). The expansion ratio was measured by the drainage method where 2.5 g of rice was placed in 50 mL of water and the elevated volume was measured. After cooking and draining, the rice was placed in 50 mL of water and the elevated volume was measured. The water absorption rate was measured by weighing 2.5 g of rice and 50 mL of water, cooking, draining, and then reweighing the rice with a filter paper (M $_{after\ cooking}$). Water absorption (%) = M $_{after\ cooking}$—2.5/ 2.5 × 100. The cooking time was determined by pressing the cooked rice grains in a hand and determining when there was no hard heart in the middle, and the time was recorded. The taste value score was measured by pressing the cooked rice into a rice cake and measuring the taste with a STA1B taste scoring apparatus (Satake, Japan). The texture characteristics (e.g., hardness, viscosity, and elasticity) were measured by pressing the cooked rice into a rice cake and measuring the textural characteristics of the rice cake using a CT34500 texture analyzer (Brookfield Engineering Laboratories). The probe (TA43 spherical probe) was set at 30 mm above the platform, the test speed was 2.5 mm/s, the test speed was 5 mm/s after the test, the trigger point load was 0.05 N, the target value was 65%, and the cycle was repeated twice. Starch pasting characteristics were measured using a rapid viscosity analyzer. A total of 3 g of milled rice flour was sampled, 25 mL of distilled water was added and this mixture was placed into an Rapid viscosity analyzer (RVA-4) (Newport Scientific, Australia) for determine the starch pasting characteristics.

Nutritional quality: protein and amylose content were determined with a fixed grating near-infrared automatic analyzer (DA7200, Perten, Sweden). The brown rice samples were tiled and filled the sample trays, and the brown rice mode was selected for automatic scanning. The crude fat was determined by Soxhlet extraction using GB5009.6–2016, with a sample of 1.00 g of brown rice flour. The starch content was determined by pretreatment of the rice sample using the acid hydrolysis method from the GB 5009.9–2016 standard, then 10 mL of the sample solution was added to 490 mL of distilled water, 0.5 mL of phenol, and 2.5 mL of concentrated sulfuric acid, and the mixture was placed in a test tube. After cooling, the

absorbance was measured at 490 nm using an ultraviolet-visible spectrophotometer (UV762 Shanghai Dingke Scientific Instrument Co., Ltd.).

### Statistical analysis

The data in all tables are the average values of three repeated observations. Data was sorted by Excel 2010 and the significant and multivariate correlation analyses were performed using the SPSS 25 software program.

## Results

### 1. Rice grain yield

The rice yields are shown in Table 1. Compared with CK, FP1, FP2, and OPT showed significant increases in rice yield, increasing by 52.0%, 64.0%, and 75.5%, respectively. However, there was no significant difference in the rice yield obtained among the three different fertilization modes. The highest yield was obtained by OPT, which was 10 040 kg·hm$^{-2}$.

### 2. Rice quality characteristics

**2.1. Rice milling quality and appearance quality.** The rice milling quality and appearance quality results are shown in Table 2. The brown rice and head rice rates of FP1 increased significantly by 5.2% and 22.8%, respectively, compared with CK. The other two fertilization modes were not significantly different from CK. FP2 had the highest milled rice rate; however, there was no significant difference in the milled rice rates among the different fertilization modes. For the rice appearance quality, the chalky kernel rate of rice after the different fertilization treatments decreased significantly compared with those obtained by CK. The three fertilization modes of FP1, FP2, and OPT decreased by 15.5%, 15.7%, and 11.8% respectively. There was no significant difference between the chalkiness. FP2 had the lowest chalky kernel rate and chalkiness.

**2.2. Rice cooking and eating quality.** The rice cooking and eating quality results are shown in Figs 1–3. There was no significant difference in the cooking quality and texture characteristics of rice treated with different fertilization (Figs 1 and 3). Compared with CK, the FP1 treatment significantly reduced the appearance score of rice (Fig 2A). The three different fertilization modes increased the hardness score of rice, and reduced the viscosity score and balance score, with significant differences among FP1, OPT, and CK (Fig 2B–2D). However, there was no significant difference between FP2 and CK (Figs 1–3). Therefore, the FP2 fertilization mode did not cause significant changes in the cooking and eating quality of rice, whereas the FP1 and OPT fertilization modes significantly reduced the eating quality.

The rice starch pasting properties results are shown in Table 3. There was no significant difference among the tested fertilization modes regarding the breakdown, setback, peak time, and

**Table 1. Effect of different fertilization modes on rice yield under the rice-crab culture system.**

| Treatment | Fertilization rate (kg·hm$^{-2}$) | Nitrogen rate (kg·hm$^{-2}$) | Yield (kg·hm$^{-2}$) |
|---|---|---|---|
| CK | 0 | 0 | 5720 ± 1869b |
| FP1 | 540 | 300 | 8693 ± 107a |
| FP2 | 379.5 | 210 | 9377 ± 821a |
| OPT | 465 | 270 | 10 040 ± 767a |

CK: no fertilization; FP1: conventional fertilization, inorganic fertilizer; FP2: one-time fertilization, organic-inorganic compound fertilizer; OPT: one-time fertilization, inorganic fertilizer. Different letters indicate statistical significance at the *P* = 0.05 level.

**Table 2. Rice milling quality and appearance quality under the rice-crab culture system.**

| Treatment | Milling quality (%) | | | Appearance quality (%) | |
|---|---|---|---|---|---|
| | Brown rice rate | Milled rice rate | Head rice rate | Chalky kernel rate | Chalkiness |
| CK | 78.44±1.11b | 78.61±3.26a | 52.11±5.05b | 44.73±5.13a | 12.62±2.52a |
| FP1 | 82.52±2.96a | 75.21±4.90a | 63.98±6.28a | 37.78±7.65b | 12.2±2.84a |
| FP2 | 78.60±0.91b | 79.54±1.82a | 58.52±4.94ab | 36.99±7.04b | 11.02±3.04a |
| OPT | 78.17±1.06b | 78.13±2.34a | 61.58±6.72ab | 39.47±2.25b | 13.29±0.90a |

CK: no fertilization; FP1: conventional fertilization, inorganic fertilizer; FP2: one-time fertilization, organic-inorganic compound fertilizer; OPT: one-time fertilization, inorganic fertilizer. Different letters indicate statistical significance at the *P* = 0.05 level.

pasting temperature. The peak, trough, and final viscosities of the four treatments had consistent results, with the FP1 treatment significantly lower than that of the CK, and was decreased by 18.0%, 17.28%, and 13.9%, respectively. Therefore, the eating quality of the rice in the FP1 mode was poor. However, there was no significant difference among FP2, OPT, and CK.

**2.3. Rice nutritional quality.** The rice nutritional quality results are shown in Fig 4. There were no significant differences in crude fat and starch content among the three fertilization

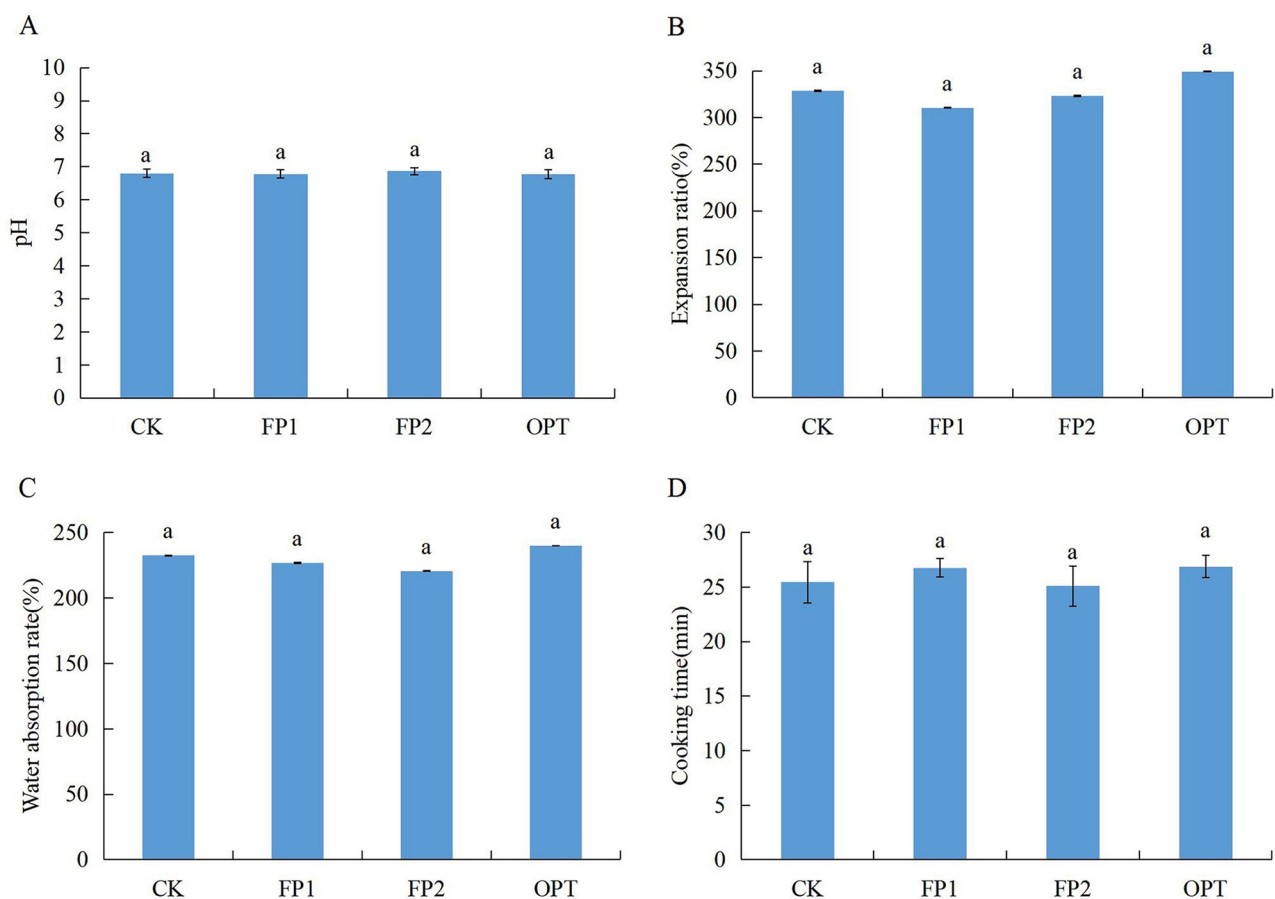

**Fig 1. Rice cooking quality under the rice-crab culture system.** CK: no fertilization; FP1: conventional fertilization, inorganic fertilizer; FP2: one-time fertilization, organic-inorganic compound fertilizer; OPT: one-time fertilization, inorganic fertilizer. Different letters indicate statistical significance at the *P* = 0.05 level.

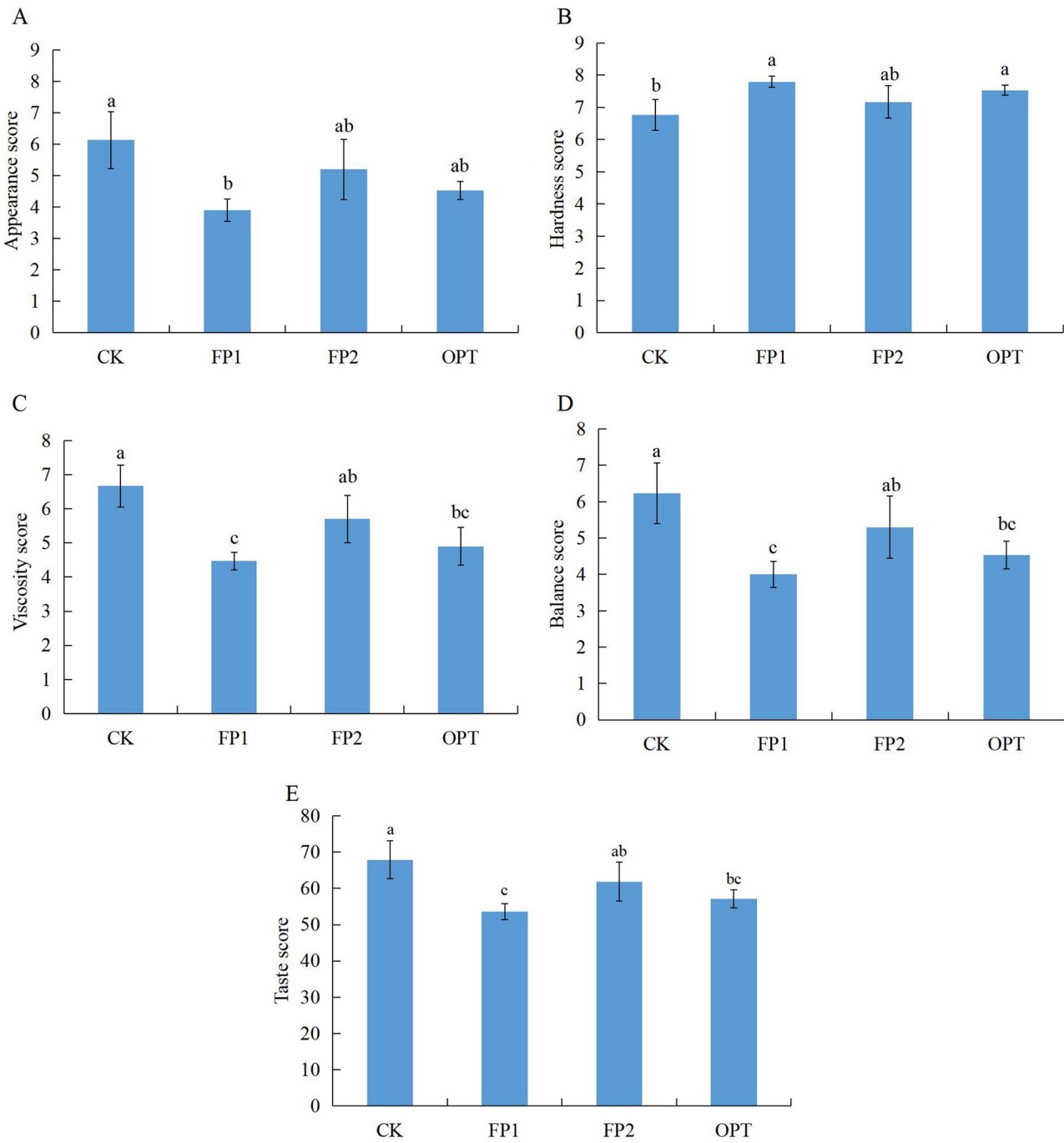

**Fig 2. Rice eating quality under the rice-crab culture system.** CK: no fertilization; FP1: conventional fertilization, inorganic fertilizer; FP2: one-time fertilization, organic-inorganic compound fertilizer; OPT: one-time fertilization, inorganic fertilizer. Different letters indicate statistical significance at the $P = 0.05$ level.

modes and CK (Fig 4B and 4C). Compared with CK, FP1, FP2, and OPT had a significantly higher protein content of 45.7%, 19.8%, and 43.3%, respectively, and FP1 and OPT were significantly higher than FP2 (Fig 4A). The three fertilization modes significantly reduced the amylose content by 3.0%, 1.4%, and 2.6%, respectively, and FP1 and OPT were significantly lower than FP2 (Fig 4D). There was no difference in starch content between rice in the different

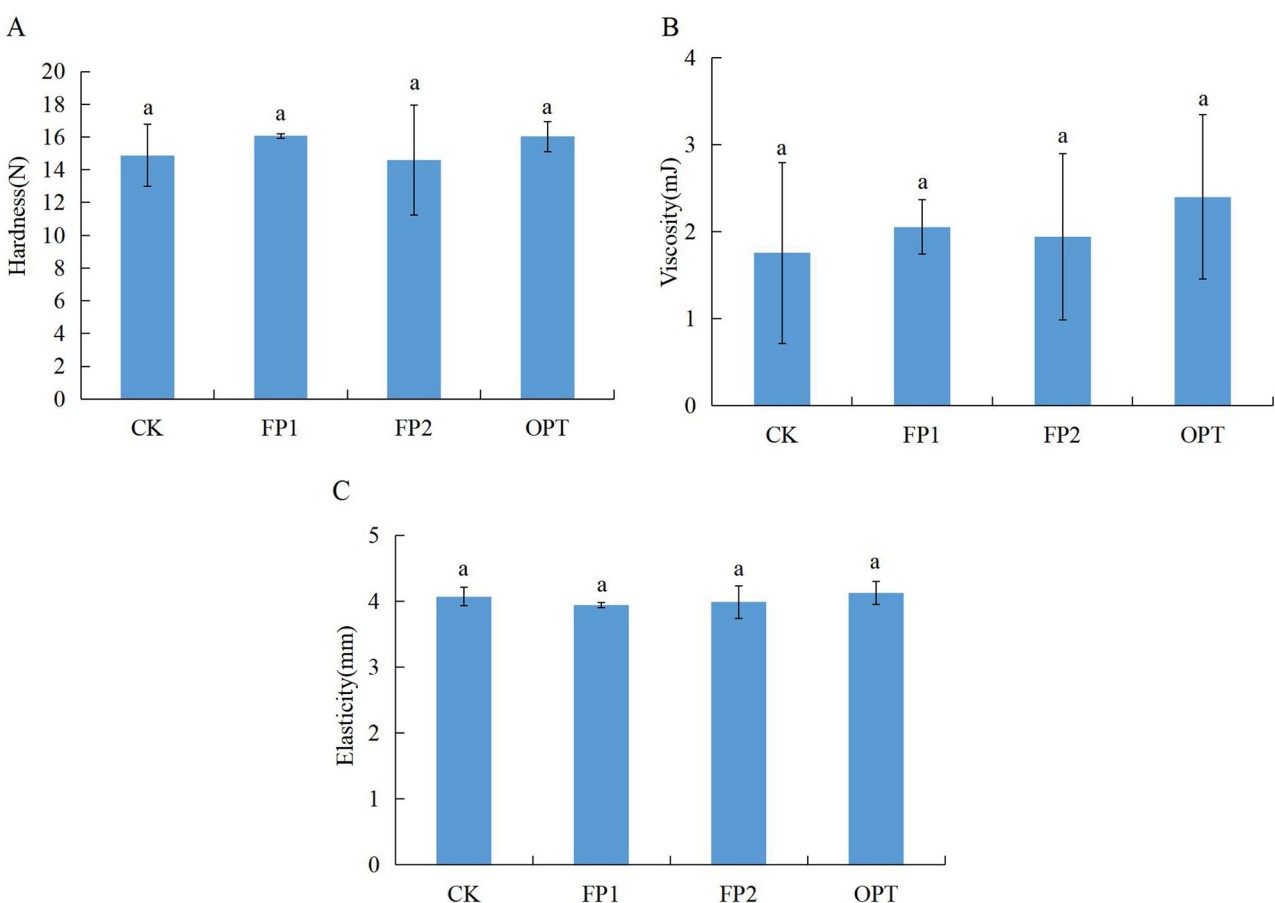

**Fig 3. Rice texture characteristics under the rice-crab culture system.** CK: no fertilization; FP1: conventional fertilization, inorganic fertilizer; FP2: one-time fertilization, organic-inorganic compound fertilizer; OPT: one-time fertilization, inorganic fertilizer. Different letters indicate statistical significance at the $P = 0.05$ level.

fertilization modes; however, there was a significant difference in amylose content, indicating that different fertilization modes may affect the type and quantity of starch synthesis. The protein content of the rice in the four modes showed a trend of FP1 > OPT > FP2 > CK, whereas the amylose trend was the opposite.

**2.4. Correlation between rice quality traits.** The correlation analysis results are shown in Table 4. The head rice rate was significantly positively correlated with protein content and

**Table 3. Rice starch pasting properties under the rice-crab culture system.**

| Treatment | Peak viscosity (RVU) | Trough viscosity (RVU) | Breakdown (RVU) | Final viscosity (RVU) | Setback(RVU) | Peak time (min) | Pasting temperature (°C) |
|---|---|---|---|---|---|---|---|
| CK | 3207±424.58a | 2089.33±161.97a | 1117.66±283.86a | 3102±270.25a | 1012.66 ±120.14a | 6.24±0.28a | 72.98±0.49a |
| FP1 | 2631±48b | 1728.33±56.5b | 902.66±8.5a | 2672±60.63b | 943.66±29.93a | 6.33±0a | 72.98±0.44a |
| FP2 | 2827.33±123.45ab | 1913±104.93ab | 914.33±80.25a | 2844.33±99.52ab | 931.33±19.73a | 6.35±0.07a | 73.21±0.49a |
| OPT | 2890.33±189.89ab | 1917±156.61ab | 973.33±67.57a | 2877.33±185.13ab | 960.33±45.54a | 6.29±0.03a | 72.65±0.05a |

CK: no fertilization; FP1: conventional fertilization, inorganic fertilizer; FP2: one-time fertilization, organic-inorganic compound fertilizer; OPT: one-time fertilization, inorganic fertilizer. Different letters indicate statistical significance at the $P = 0.05$ level.

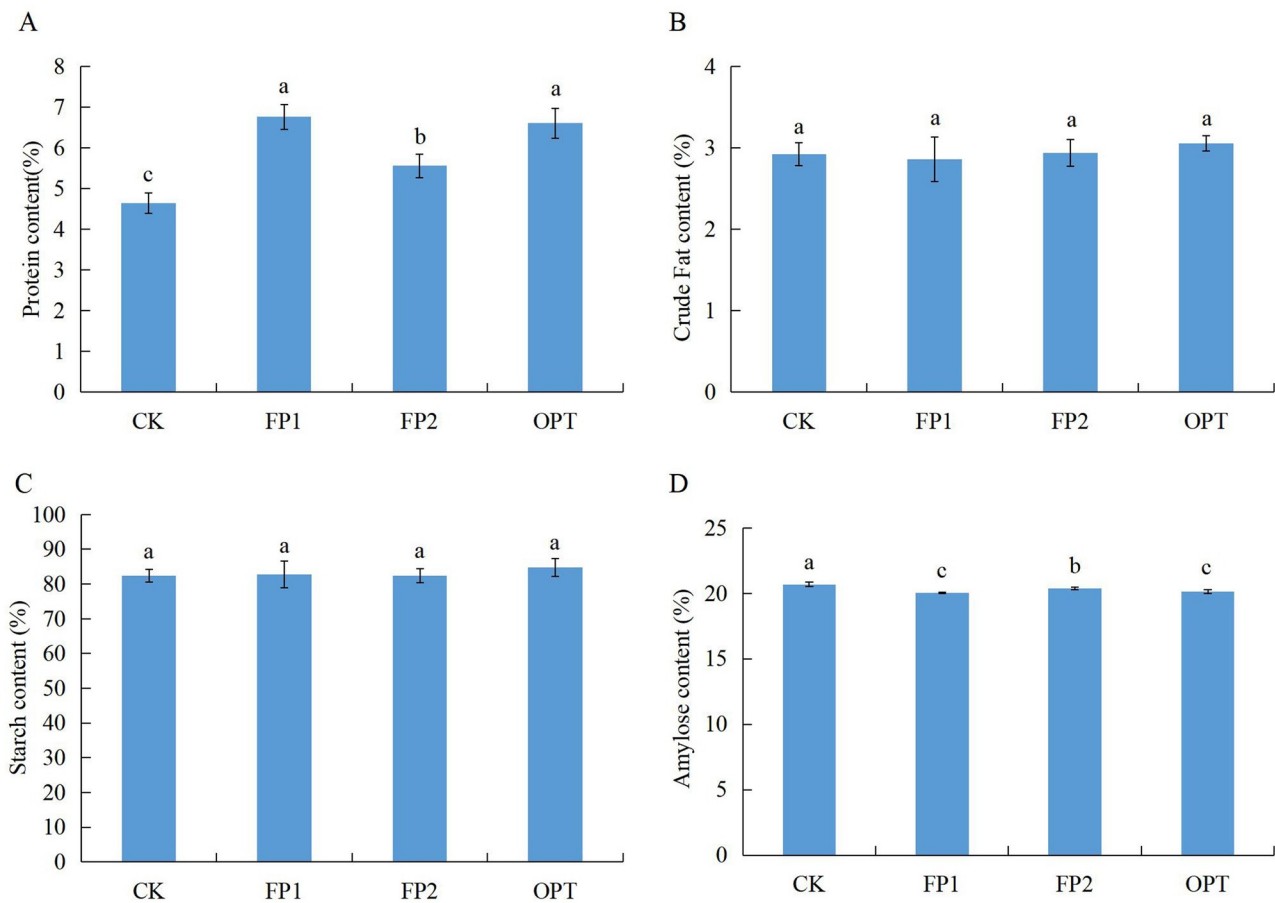

**Fig 4. Rice nutritional quality under the rice-crab culture system.** CK: no fertilization; FP1: conventional fertilization, inorganic fertilizer; FP2: one-time fertilization, organic-inorganic compound fertilizer; OPT: one-time fertilization, inorganic fertilizer. Different letters indicate statistical significance at the $P = 0.05$ level.

hardness score; however, it was negatively correlated with appearance, viscosity, balance, and taste scores, and other cooking and eating quality and amylose content. The chalkiness was significantly positively correlated with the water absorption rate. There was a significant negative correlation between protein content and appearance, viscosity, balance, and taste scores, and other cooking and eating quality, and a significant positive correlation with hardness score. Amylose was the opposite of protein and there was a significant negative correlation between them.

## Discussion

### 1. Yield

Rice yield is the result of a combination of rice genetics, various environmental factors, fertilization modes, and nitrogen application rates; thus, the factors affecting yield are diverse and complex [16]. The results of the different fertilization modes on rice yield of rice-crab culture system obtained from previous studies differ to each other. Some researchers believed that the rice-crab culture system would increase rice yield [17,18]. This might be due to the additional nutrients from crab feces, crab movements accelerating the release of accumulated nutrients, and crab feeding on aquatic plants contributing to nutrient recycling and decreasing loss of N

**Table 4. Correlation between rice quality traits under the rice-crab culture system.**

| | Brown rice rate | Milled rice rate | Head rice rate | Chalky kernel rate | Chalkiness | pH | Expansion ratio | Water absorption rate | Cooking time | Appearance score | Hardness score | Viscosity score | Balance score | Taste score | Hardness | Viscosity | Elastic | Peak viscosity | Trough viscosity | Breakdown | Final viscosity | Setback | Peak time | Pasting temperature | Protein content | Fat content | Starch content | Amylose content |
|---|---|---|---|---|---|---|---|---|---|---|---|---|---|---|---|---|---|---|---|---|---|---|---|---|---|---|---|---|
| Brown rice rate | 1 | | | | | | | | | | | | | | | | | | | | | | | | | | | |
| Milled rice rate | -0.921 | 1 | | | | | | | | | | | | | | | | | | | | | | | | | | |
| Head rice rate | 0.608 | -0.642 | 1 | | | | | | | | | | | | | | | | | | | | | | | | | |
| Chalky kernel rate | -0.388 | 0.217 | -0.811 | 1 | | | | | | | | | | | | | | | | | | | | | | | | |
| Chalkiness | -0.137 | -0.257 | 0.032 | 0.506 | 1 | | | | | | | | | | | | | | | | | | | | | | | |
| pH | -0.295 | 0.644 | -0.332 | -0.280 | -0.904 | 1 | | | | | | | | | | | | | | | | | | | | | | |
| Expansion ratio | -0.779 | 0.498 | -0.140 | 0.263 | 0.624 | -0.287 | 1 | | | | | | | | | | | | | | | | | | | | | |
| Water absorption rate | -0.336 | -0.058 | -0.027 | 0.490 | 0.975* | -0.800 | 0.782 | 1 | | | | | | | | | | | | | | | | | | | | |
| Cooking time | 0.457 | -0.732 | 0.744 | -0.222 | 0.677 | -0.873 | 0.202 | 0.589 | 1 | | | | | | | | | | | | | | | | | | | |
| Appearance score | -0.691 | 0.743 | -.990* | 0.739 | -0.093 | 0.422 | 0.197 | -0.006 | -0.794 | 1 | | | | | | | | | | | | | | | | | | |
| Hardness score | 0.681 | -0.750 | -.986* | -0.713 | 0.137 | -0.460 | -0.168 | 0.049 | 0.821 | -.999** | 1 | | | | | | | | | | | | | | | | | |
| Viscosity score | -0.625 | 0.697 | -.993** | 0.738 | -0.136 | 0.438 | 0.110 | -0.065 | -0.814 | .996** | -.997** | 1 | | | | | | | | | | | | | | | | |
| Balance score | -0.659 | 0.726 | -.991** | 0.730 | -0.128 | 0.443 | 0.149 | -0.047 | -0.813 | .999** | -.999** | .999** | 1 | | | | | | | | | | | | | | | |
| Taste score | -0.662 | 0.725 | -.992** | 0.735 | -0.119 | 0.436 | 0.155 | -0.038 | -0.808 | .999** | -.999** | .999** | 1.000** | 1 | | | | | | | | | | | | | | |
| Hardness | 0.517 | -0.779 | 0.751 | -0.225 | 0.652 | -0.872 | 0.134 | 0.549 | .997** | -0.808 | 0.833 | -0.821 | -0.823 | -0.819 | 1 | | | | | | | | | | | | | |
| Viscosity | -0.019 | -0.221 | 0.723 | -0.447 | 0.509 | -0.527 | 0.580 | 0.560 | 0.788 | -0.683 | 0.703 | -0.745 | -0.718 | -0.713 | 0.748 | 1 | | | | | | | | | | | | |
| Elastic | -0.787 | 0.483 | -0.345 | 0.522 | 0.714 | -0.356 | .960* | 0.846 | 0.137 | 0.372 | -0.338 | 0.295 | 0.327 | 0.334 | 0.076 | 0.396 | 1 | | | | | | | | | | | |
| Peak viscosity | -0.717 | 0.611 | -0.934 | 0.906 | 0.304 | 0.049 | 0.436 | 0.378 | -0.494 | 0.920 | -0.902 | 0.899 | 0.905 | 0.909 | -0.517 | -0.452 | 0.638 | 1 | | | | | | | | | | |
| Trough viscosity | -0.821 | 0.753 | -0.935 | 0.806 | 0.187 | 0.201 | 0.482 | 0.297 | -0.593 | 0.948 | -0.935 | 0.920 | 0.932 | 0.935 | -0.623 | -0.433 | 0.647 | .981* | 1 | | | | | | | | | |
| Breakdown | -0.510 | 0.354 | -0.867 | .990* | 0.457 | -0.181 | 0.335 | 0.471 | -0.312 | 0.813 | -0.788 | 0.805 | 0.801 | 0.806 | -0.322 | -0.449 | 0.580 | .957* | 0.882 | 1 | | | | | | | | |
| Final viscosity | -0.757 | 0.663 | -0.938 | 0.874 | 0.266 | 0.101 | 0.456 | 0.353 | -0.529 | 0.933 | -0.917 | 0.909 | 0.918 | 0.921 | -0.555 | -0.445 | 0.646 | .998** | .992** | 0.935 | 1 | | | | | | | |
| Setback | -0.357 | 0.172 | -0.779 | .999** | 0.543 | -0.329 | 0.265 | 0.519 | -0.172 | 0.703 | -0.675 | 0.702 | 0.693 | 0.698 | -0.174 | -0.417 | 0.524 | 0.885 | 0.778 | .982* | 0.851 | 1 | | | | | | |
| Peak time | 0.409 | -0.159 | 0.685 | -.972* | -0.690 | 0.455 | -0.428 | -0.682 | 0.023 | -0.615 | 0.581 | -0.599 | -0.596 | -0.602 | 0.035 | 0.224 | -0.662 | -0.859 | -0.752 | -.957* | -0.826 | -.979* | 1 | | | | | |
| Pasting temperature | 0.158 | 0.231 | -0.270 | -0.226 | -0.945 | 0.859 | -0.723 | -.955* | -0.782 | 0.296 | -0.337 | 0.355 | 0.336 | 0.328 | -0.745 | -0.760 | -0.721 | -0.089 | -0.014 | -0.193 | -0.065 | -0.263 | 0.448 | 1 | | | | |
| Protein content | 0.536 | -0.648 | -.980* | -0.698 | 0.229 | -0.491 | 0.015 | 0.174 | 0.853 | -.977* | .983* | -.992** | -.986* | -.985* | 0.852 | 0.822 | -0.172 | -0.845 | -0.863 | -0.758 | -0.854 | -0.661 | 0.536 | -0.457 | 1 | | | |
| Fat content | -0.745 | 0.509 | 0.019 | 0.033 | 0.488 | -0.180 | .971* | 0.662 | 0.213 | 0.063 | -0.040 | -0.028 | 0.015 | 0.021 | 0.142 | 0.679 | 0.867 | 0.254 | 0.333 | 0.116 | 0.285 | 0.032 | -0.201 | -0.657 | 0.147 | 1 | | |
| Starch content | -0.080 | -0.238 | 0.575 | -0.188 | 0.726 | -0.702 | 0.678 | 0.764 | 0.835 | -0.561 | 0.591 | -0.626 | -0.600 | -0.594 | 0.794 | .961* | 0.560 | -0.250 | -0.269 | -0.204 | -0.256 | -0.154 | -0.049 | -0.910 | 0.718 | 0.713 | 1 | |
| Amylose content | -0.583 | 0.665 | -.992** | 0.739 | -0.156 | 0.440 | 0.060 | -0.094 | -0.819 | .990* | -.992** | .999** | .995** | .995** | -0.823 | -0.778 | 0.251 | 0.886 | 0.901 | 0.800 | 0.894 | 0.703 | -0.591 | 0.384 | -.997** | -0.080 | -0.659 | 1 |

* , significant at the P = 0.05 level;

** , significant at the P = 0.01 level.

[14]. However, other scholars believed that the presence of crabs had no significant effect on rice yield [19]. In the present study, the rice yield in the crab-free system that had the same variety and planting environment and the same fertilization mode and amount as that found in FP1 was measured. The yield of rice in the crab-free system was 11 017 kg·hm$^{-2}$ (data not shown), which was slightly higher than that in the rice-crab culture system; however, there was no significant difference in the yield difference among the three fertilization modes. This might be due to the long-term maintenance of the deep-water layer, which is not conducive to rice root growth and tillering, or because the crabs crawl in the field, causing damage to the rice tillers and decreasing the number of effective panicles [20].

In the rice-crab culture system, the main factors affecting rice yield are the fertilization mode and nitrogen application rate. After fertilization, the rice yield increased significantly. However, this increase was not proportional to the amount of nitrogen applied. The fertilizer application rate of the traditional fertilization mode, FP1, was too high, which not only reduces rice yield but also causes soil pollution. In the traditional fertilization mode, the application amount of nitrogen, phosphorus, and potassium was unreasonable and the application of the base fertilizer was emphasized. Studies have shown that this will make rice grow too strong in the early growth stage, reducing effective tillering and increasing ineffective tillering, which was the possible cause of the decline in yield [21]. Jia et al. [22] also studied the performance of Yanfeng 47 rice variety and found that the application of nitrogen fertilizer increased the number of tillers for Yanfeng 47 resulted in the yield. However, excessive application of nitrogen fertilizer leads to delay or immature rice growth. Therefore, reasonable control of the application rate of nitrogen fertilizer creates high rice yield. The nitrogen application rate of FP2 was 22.2% lower than that of OPT and yet the rice yield was only decreased by 6.6% and there was no significant difference from that of OPT. This might be due to the application of organic fertilizer to FP2 for increasing the rice yield [23]. The combination of organic and inorganic fertilizers can promote nutrient synchronization and nitrogen use efficiency and reduce nutrient loss by converting inorganic nitrogen into organic forms [24]. An et al. [25] found that when the external environmental factors were good, the appropriate application of organic fertilizer increased rice yield under the rice-crab culture system, which was close to or even higher than simply planting rice. Walia et al. [26] also showed that the combination of organic and chemical fertilizers increased the 1000-grain weight, protein content, and quality parameters of rice compared with chemical fertilizer alone. In the present study, the FP2 mode showed lower economic costs and higher rice yield. Therefore, in terms of yield, the FP2 application of organic-inorganic compound fertilizer was more suitable for rice production in the rice-crab culture system.

## 2. Grain quality characteristics

As the main measure of rice quality, the head rice rate reflects the milling quality and commodity value of rice milling [27]. With an increase in the nitrogen application rate, the brown rice and head rice rates of FP1 were significantly improved. Correlation between rice quality showed that the head rice rate was significantly positively correlated with the protein content and hardness score, with correlation coefficients of 0.980 and 0.986, respectively; however, it was negatively correlated with the appearance, viscosity, balance, and taste scores, and other cooking and eating quality and amylose content. Therefore, the better the quality of the milling, the worse the cooking and eating quality of the rice. This was mainly due to the increase of protein content and decrease of amylose content in rice, as well as the increase of rice hardness and the ability of rice to resist grinding damage. However, palatability was reduced.

Chalkiness is a white opaque area produced by the loose accumulation of starch granules in rice grains and is an unfavorable feature of the appearance quality of rice grains [28]. Chalkiness also reduces the milling and cooking and eating quality of rice [29]. Therefore, rice production should aim to reduce the chalkiness of rice. The formation of rice chalkiness is significantly affected by rice varieties and environmental factors such as nitrogen application and temperature [30]. In the present study, the application of nitrogen fertilizer significantly reduced the chalky kernel rate of rice. This was due to the application of nitrogen fertilizer to cause protein accumulation in rice resulting in more space occupied by protein bodies, dense starch particles, and improved appearance quality [31]. Qiao et al. [32] also showed that nitrogen fertilizer had a good effect on decreasing the chalky kernel rate. However, with a further increase in the nitrogen level, the total amount of spikelets increased, and the grain filling period was enlarged resulting in lower grain filling rate, poor grain filling, and increase in false grains. In this case, the arrangement of the starch granules was loose resulting in chalkiness of the rice [33]. Chalkiness was significantly positively correlated with the water absorption rate in the present study ($R^2 = 0.975$). The higher whiteness indicated that the starch granules in the rice were loosely arranged and the voids were large. The starch easily swelled during cooking; therefore, the water absorption rate was high.

In terms of cooking and eating quality, rice produced in the FP1 fertilization mode was the worst and FP2 was the best. Starch is the main component in rice and starch pasting significantly affects the texture of rice [34]. The viscosity analysis results showed that the peak, trough, and final viscosities of FP1 were significantly lower than that of CK; however, there was no significant difference between FP2 and CK. According to the correlation results, the rice taste score was positively correlated with the starch pasting viscosity, although the correlation was not significant. However, it was enough to prove that the cooking quality of the FP1 rice was poor.

Protein content can significantly affect the cooking and eating quality of rice [35]. The results of the present study indicated that the protein content increased as the nitrogen application rate increased in the rice-crab culture system. There was a significant negative correlation between protein content and appearance, viscosity, balance, and taste scores, and other cooking taste quality ($R^2 = -0.977, -0.992, -0.986, -0.985$, respectively) and a significant positive correlation with hardness score ($R^2 = 0.983$). This was in agreeance with the results of the cooking and eating quality measurements. Nitrogen application resulted in a significant increase in protein content in rice and a decrease in cooking and eating quality. The lowest protein content of FP2 had the highest taste score and the rice had the best cooking and eating quality. This might be due to the network structure formed by the disulfide bond of the protein to affect the starch pasting, making the rice texture harder and less viscous, resulting in a poor taste quality of the rice [36]. Baxter et al. [37] showed that the most abundant gluten in rice protein interacted directly with the starch granules, blocking the entry of water molecules and resulting in slower water absorption of the starch granules and lower water absorption rate. This effect was concentration dependent and was consistent with the results of the present study. Protein content was a key factor in terms of milling, appearance, and cooking and eating qualities. However, the application rate of nitrogen in each mode was not the same and the nitrogen content in the field was complicated towing to the residues of crab feed and the excretion of crab manure. The mechanism of the effect on rice quality requires further evaluation.

Starch includes two types of molecules: amylose and amylopectin. Most researchers believe that the amylose and protein content in rice is the most important chemical component determining the cooking and eating quality of rice [38]. The mechanism of the effect of starch on rice texture must be further studied. In the present study, amylose content decreased as the amount of nitrogen applied in the four fertilization modes increased. This might be because

the application of nitrogen fertilizer delayed the accumulation of starch during the early stage of grain filling and significantly reduced the amylose content. Rice containing lower amylose has better softness and palatability and is preferred by consumers. However, rice with higher amylose content tends to have a harder and less sticky texture after cooking [39]. This is due to the amylose molecules limiting starch swelling and leaching during cooking by entanglement and/or co-crystallization with amylopectin, thus affecting the viscosity between cooked rice grains [40] and resulting in a harder texture, limited starch leaching, reduced eating quality of rice [41,42]. Although the application of nitrogen fertilizer reduced the amylose content and improved the cooking and eating quality, excessive nitrogen application significantly increased the protein content and ultimately reduced the cooking and eating quality of rice. This was consistent with the results of the correlation analysis, where there was a significant positive correlation between amylose and appearance, viscosity, balance, and taste scores, and other cooking and eating quality ($R^2$ = 0.990, 0.999, 0.995, 0.995, respectively), and negatively correlated with hardness score ($R^2$ = -0.992). In summary, reasonable fertilization was the key to improving rice quality.

## Conclusions

In the rice-crab culture system, the three fertilization modes significantly increased the rice yield compared with the control, but there was no significant difference in the yield under the three modes. FP1 treatment improved the milling quality and FP2 treatment improved the appearance quality. Out of the three treatments, FP2 treatment maintained good eating quality. From the perspective of nutritional quality, the protein content of the FP2 treatment was reasonable, although it was not the highest, but it contributed to the formation of a good comprehensive quality of rice. In addition, FP2 treatment reduced the amount of nitrogen fertilizer applied compared with traditional fertilization mode, the cost was reduced and the yield could be guaranteed. The mode of FP2 could be recommended as proper fertilization mode for the rice-crab culture system in Panjin area.

The rice-crab culture model has been proven to improve the ecological environment of rice fields and increase economic benefits. In the future, focus should be raised to improve modern rice cultivation management techniques and rice-crab culture techniques and adopting a reasonable fertilization mode to further improve the yield and quality of rice.

## Author Contributions

**Conceptualization:** Wanning Zhao, Zhaoxia Wu, Wentao Sun.

**Data curation:** Wanning Zhao, Hanling Liang, Yu Fu, Yubo Liu, Chen Yang, Tian Zhang, Tianyu Wang, Liyan Rong, Shuang Zhang, Zhaoxia Wu.

**Formal analysis:** Wanning Zhao.

**Funding acquisition:** Zhaoxia Wu, Wentao Sun.

**Investigation:** Wanning Zhao, Zhaoxia Wu.

**Methodology:** Wanning Zhao, Hanling Liang, Yu Fu, Yubo Liu, Chen Yang, Tian Zhang, Tianyu Wang, Liyan Rong, Shuang Zhang, Zhaoxia Wu.

**Project administration:** Wanning Zhao, Hanling Liang, Zhaoxia Wu, Wentao Sun.

**Resources:** Wanning Zhao, Zhaoxia Wu, Wentao Sun.

**Software:** Wanning Zhao.

**Supervision:** Wanning Zhao, Hanling Liang, Yu Fu, Yubo Liu, Chen Yang, Tian Zhang, Tianyu Wang, Liyan Rong, Shuang Zhang, Zhaoxia Wu.

**Validation:** Wanning Zhao, Hanling Liang, Yu Fu, Yubo Liu, Chen Yang, Tian Zhang, Tianyu Wang, Liyan Rong, Shuang Zhang, Zhaoxia Wu, Wentao Sun.

**Visualization:** Wanning Zhao, Hanling Liang, Yu Fu, Yubo Liu, Chen Yang, Tian Zhang, Tianyu Wang, Liyan Rong, Shuang Zhang.

**Writing – original draft:** Wanning Zhao.

**Writing – review & editing:** Wanning Zhao, Zhaoxia Wu.

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
