## [Decision Letter · Decision Letter 0]

10 Feb 2020

PONE-D-20-01018

Effects of different fertilization modes on rice yield and quality under a rice-crab culture system

PLOS ONE

Dear Mrs. Wu,

Thank you for submitting your manuscript to PLOS ONE. After careful consideration, we feel that it has merit but does not fully meet PLOS ONE’s publication criteria as it currently stands. Therefore, we invite you to submit a revised version of the manuscript that addresses the points raised during the review process.

We would appreciate receiving your revised manuscript by Mar 26 2020 11:59PM. To enhance the reproducibility of your results, we recommend that if applicable you deposit your laboratory protocols in protocols.io, where a protocol can be assigned its own identifier (DOI) such that it can be cited independently in the future. For instructions see: http://journals.plos.org/plosone/s/submission-guidelines#loc-laboratory-protocols

We look forward to receiving your revised manuscript.

Kind regards,

Vassilis G. Aschonitis

Academic Editor

PLOS ONE

Journal Requirements:

1. PLOS requires an ORCID iD for the corresponding author in Editorial Manager on papers submitted after December 6th, 2016. Please ensure that you have an ORCID iD and that it is validated in Editorial Manager. To do this, go to ‘Update my Information’ (in the upper left-hand corner of the main menu), and click on the Fetch/Validate link next to the ORCID field. This will take you to the ORCID site and allow you to create a new iD or authenticate a pre-existing iD in Editorial Manager. Please see the following video for instructions on linking an ORCID iD to your Editorial Manager account: https://www.youtube.com/watch?v=_xcclfuvtxQ

Reviewers' comments:

Reviewer's Responses to Questions

**Comments to the Author**

1. Is the manuscript technically sound, and do the data support the conclusions?

Reviewer #1: Yes

2. Has the statistical analysis been performed appropriately and rigorously? 

Reviewer #1: Yes

3. Have the authors made all data underlying the findings in their manuscript fully available?

Reviewer #1: Yes

4. Is the manuscript presented in an intelligible fashion and written in standard English?

Reviewer #1: Yes

5. Review Comments to the Author

Reviewer #1: plz consider the minor corrections have been inserted in ur manuscript in the attach file.

plz try to rewrite the abstract more clear than the current one, since it is has to all treatment.

plz see the attach file

6. PLOS authors have the option to publish the peer review history of their article (what does this mean?). If published, this will include your full peer review and any attached files.

Reviewer #1: No

---

## [Author Response · Author response to Decision Letter 0]

29 Feb 2020

1.Is the manuscript technically sound, and do the data support the conclusions?

Reviewer #1: Yes

Responses to Questions:

The manuscript describes a scientifically sound study. This experiment investigated the effects of three different fertilization modes on rice yield and quality in a rice-crab culture system in Panjin, China. All indicators were tested in accordance with national standards or recognized scientific methods. The study was conducted in the field using a randomized complete block design with three replicates, including three non-fertilized control groups, for a total of 12 rice samples. Special personnel were responsible for the rice planting, fertilization, and harvesting processes, and the operation was rigorous and standard. The conclusions were based on statistical analysis of the data, and are consistent with the results of the one-year preliminary experiment. 

2. Has the statistical analysis been performed appropriately and rigorously? 

Reviewer #1: Yes

Responses to Questions:

The data measured in this test were collated by Excel 2010, and the average and standard deviation of three repeated tests for each index were calculated. SPSS 25 software was used to perform a one-way ANOVA test using the LSD method and Duncan method, with a significance level of 0.05, and a Pearson correlation coefficient was used for bivariate correlation analysis.

3. Have the authors made all data underlying the findings in their manuscript fully available?

Reviewer #1: Yes

Responses to Questions:

In our analysis, we have made full use of all the data in the manuscript to reach the most accurate and reliable conclusions. All the data on which the figures in the manuscript are based, including the mean, standard deviation, and significant differences, are stored in a public repository recommended by PLOS– figshare. The data involved are data for all indicators of rice cooking and eating, and nutritional quality. The data listed in the tables are real data.

4.Is the manuscript presented in an intelligible fashion and written in standard English?

Reviewer #1: Yes

Responses to Questions: 

The manuscript has been written in standard English. The author has corrected the grammatical errors pointed out by the reviewers and has carefully checked them.

5. Review Comments to the Author

Reviewer #1: plz consider the minor corrections have been inserted in ur manuscript in the attach file.

plz try to rewrite the abstract more clear than the current one, since it is has to all treatment.

plz see the attach file

Responses to Questions: 

Thank you very much for the excellent suggestions listed in our manuscript. We confirm that this work has not been published elsewhere and that all listed authors have approved the manuscript. All rice samples used in the test were taken from the rice experimental demonstration base of the national R&D project. Therefore, this manuscript will not encounter issues such as dual publishing and contravention of research ethics.

We are very grateful to the reviewers for their valuable comments. The author has adopted and revised the text according to the feedback and made appropriate supplementary amendments to the manuscript. 

In response to your questions about the proportion of organic and inorganic fertilizers in the FP2 fertilization mode, we added an explanation in the Materials and Methods section as follows. “The ratio of inorganic nitrogen to organic nitrogen was 7:3, inorganic nitrogen provided 70% of the nitrogen in the mixed fertilizer and organic nitrogen provided 30% of the nitrogen. Chicken manure was used as organic fertilizer, and the nitrogen content was determined in advance.”

Regarding your questions about high available nitrogen and low available potassium, we offer the following explanation. The Panjin area has coastal saline-alkali soil. We have tested the available nitrogen content in the soil at 85–120 mg/kg for many years. The effective nitrogen content determined by this test was 105.2 mg/kg, which is normal. Regarding potassium, our test results were indeed low, which may be related to the soil conditions of the test field.

Finally, we modified the abstract and deleted the redundant description of the three different fertilization modes because their specific descriptions are included in the materials section, which makes the abstract clearer and more concise.

6.PLOS authors have the option to publish the peer review history of their article (what does this mean?). If published, this will include your full peer review and any attached files.

Do you want your identity to be public for this peer review? For information about this choice, including consent withdrawal, please see our Privacy Policy.

Reviewer #1: No

Responses to Questions:

Please do not reveal my identity for this peer review. I would prefer to remain anonymous. Thank you.

---

## [Editor Report · Decision Letter 1]

4 Mar 2020

Effects of different fertilization modes on rice yield and quality under a rice-crab culture system

PONE-D-20-01018R1

Dear Dr. Wu,

We are pleased to inform you that your manuscript has been judged scientifically suitable for publication and will be formally accepted for publication once it complies with all outstanding technical requirements.

With kind regards,

Vassilis G. Aschonitis

Academic Editor

PLOS ONE
---

## [Editor Report · Acceptance letter]

11 Mar 2020

PONE-D-20-01018R1 

Effects of different fertilization modes on rice yield and quality under a rice-crab culture system 

Dear Dr. Wu:

I am pleased to inform you that your manuscript has been deemed suitable for publication in PLOS ONE. Congratulations! Your manuscript is now with our production department. 

With kind regards,

on behalf of

Dr. Vassilis G. Aschonitis 

Academic Editor

PLOS ONE